# Histamine Intolerance Originates in the Gut

**DOI:** 10.3390/nu13041262

**Published:** 2021-04-12

**Authors:** Wolfgang J. Schnedl, Dietmar Enko

**Affiliations:** 1General Internal Medicine Practice, Dr. Theodor Körnerstrasse 19b, A-8600 Bruck, Austria; 2Clinical Institute of Medical and Chemical Laboratory Diagnostics, Medical University of Graz, Auenbruggerplatz 15, A-8036 Graz, Austria; enko.dietmar@gmx.at

**Keywords:** histamine, diamine oxidase, gastrointestinal, biogenic amines, food intolerance, food malabsorption

## Abstract

Histamine intolerance (HIT) is assumed to be due to a deficiency of the gastrointestinal (GI) enzyme diamine oxidase (DAO) and, therefore, the food component histamine not being degraded and/or absorbed properly within the GI tract. Involvement of the GI mucosa in various disorders and diseases, several with unknown origin, and the effects of some medications seem to reduce gastrointestinal DAO activity. HIT causes variable, functional, nonspecific, non-allergic GI and extra-intestinal complaints. Usually, evaluation for HIT is not included in differential diagnoses of patients with unexplained, functional GI complaints or in the here-listed disorders and diseases. The clinical diagnosis of HIT is challenging, and the thorough anamnesis of all HIT-linked complaints, using a standardized questionnaire, is the mainstay of HIT diagnosis. So far, DAO values in serum have not been established to correlate with DAO activity in the gut, but the diagnosis of HIT may be supported with determination of a low serum DAO value. A targeted dietary intervention, consisting of a histamine-reduced diet and/or supplementation with oral DAO capsules, is helpful to reduce HIT-related symptoms. This manuscript will present why histamine should also be taken into account in the differential diagnoses of patients with various diseases and disorders of unknown origin, but with association to functional gastrointestinal complaints. In this review, we discuss currently increasing evidence that HIT is primarily a gastrointestinal disorder and that it originates in the gut.

## 1. Introduction

Chronic and unexplained, functional, gastrointestinal (GI) symptoms impact more than 20% of the population. In histamine intolerance (HIT), the impairment of GI histamine degradation causes functional, nonspecific, non-allergic GI complaints and extra-intestinal complaints [1]. An unbalanced and elevated quantity of histamine in HIT seems to be the main consequence of the ingestion of histamine-containing foods [2]. Predominantly, in HIT, the intestinal enzyme diamine oxidase (DAO) has a reduced ability to metabolize and degrade histamine. Scientific evidence and studies to support this idea are increasing [3]. However, in association with various different disorders and diseases listed here, several with unknown underlying pathophysiologic mechanisms, the majority reports on additional unexplained, functional GI symptoms. Nonetheless, reliable standardized evaluations and/or laboratory tests for a definite diagnosis of HIT are still needed. HIT requires detailed anamnesis and diagnostic examination with available tests [4]. Taking into account the possible presence of additional GI diseases or disorders, the evaluation of all etiologic- and symptoms-causing factors is essential [5]. Subsequently, a personalized treatment with the targeted dietary intervention for each patient with HIT, using a histamine-reduced diet and/or oral supplementation with DAO, may help to provide sustained relief [6].

Here we review various disorders and diseases with unexplained, functional, GI complaints, and their parallels to HIT. We describe emerging evidence, and with help of published studies, report on the link between reduced gastrointestinal DAO activity and HIT, and why this association demonstrates that HIT originates in the gut.

## 2. Histamine

Histamine [2-(4-imidazolyl)-ethylamine] is included in a group of biogenic amines with putrescine, cadaverine, and tyramine, amongst others, produced by bacterial fermentation [7]. The decarboxylation of the amino acid histidine results in histamine. Generally, histamine was discovered more than 100 years ago. The understanding of the central role of histamine in allergies and in immune regulations has led to the development of antihistamine medications [8].

A disproportionate amount of histamine in the body is suspected to result from the consumption of histamine-containing foods or drinks, and the reduced ability of enzymes to digest and degrade histamine. In foods, the manufacturing process, the cleanliness of materials, the microbial composition, and the fermentation influence the amount of histamine contained. The European Union allows the histamine content in food up to a maximum of 200 mg/kg in fresh fish and 400 mg/kg in seafood products [9]. Histamine contents of more than 40 mg per meal (0.75 mg/kg body weight) considerably increase the risk of scombroid poisoning [7]. However, this intoxication is in most cases caused by the consumption of decaying sea animals, which are contaminated with bacteria that cause high concentrations of histamine. Scombroid poisoning may also arise due to spoiled food because cooking, smoking, or freezing does influence [10], but not eliminate histamine. Usually, the consumption of low amounts of histamine and biogenic amines does not cause health problems in humans. Nonetheless, HIT is clearly separated from food allergies, like, e.g., peanut allergy [11], and is described as a non-allergic adverse reaction to ingested food [3].

## 3. Histamine Intolerance (HIT)

The term “histamine intolerance” is used similarly to lactose intolerance (LIT). LIT, with a deficiency of the enzyme lactase, shows parallels to the definition of HIT, with a deficiency of the GI enzyme diamine oxidase (DAO). Occasionally, HIT is also defined as “enteral histaminosis” or “sensitivity to dietary histamine” [2]. Although, HIT has also been suggested to be a metabolic disease [12], the inadequate digestion seems to cause excess histamine throughout the body that initiates a wide variety of symptoms. Nevertheless, histamine is also involved in the etiology of many common diseases [13].

Genetic expression of DAO is mainly in the small intestine, the ascending colon, the placenta, and the kidneys [14,15]. DAO activity, as shown in rats’ intestines, increases from the duodenum to the ileum and is located in the intestinal villi [16]. It is synthesized by mature intestinal enterocytes and is constantly released from the intestinal mucosa into the gut as well as into the blood circulation, during digestion [17,18]. Ingested histamine, clearly below the dose of scombroid poisoning [19], is alleged to cause HIT-related symptoms. Human DAO and histamine N-methyl transferase, extra- and intracellular, respectively, are enzymes which catalyse the oxidative deamination of mono-, di-, and polyamines. Within the GI tract, DAO appears as the primary enzyme responsible for degrading ingested or microbiota-generated histamine [20,21].

The clinical diagnosis of HIT is still challenging, as standardized diagnostic tests are lacking. The thorough anamnesis of all HIT-linked complaints is the mainstay of HIT diagnosis. So far, a HIT questionnaire for volunteers, who randomly attended an outpatient clinic, has not proven useful for detection of HIT [22]. However, to evaluate patients with unexplained functional, nonspecific, non-allergic GI and extra-intestinal symptoms, suspected to have HIT, a questionnaire as shown in Table 1 may be helpful. We published a survey which listed 23 HIT symptoms in four categories including gastrointestinal, cardiovascular, respiratory, and skin complaints. This questionnaire contains questions about symptoms and the severity of them that patients experience due to ingestion of foods, and it is based on known symptoms related to the four histamine receptors [23]. In this description the most common and most severe symptom indicated by HIT patients in more than 90% was bloating. Other very commonly related GI symptoms included: postprandial fullness and diarrhea (both in >70% of patients), abdominal pain (>65%), and constipation (55%) [1]. This questionnaire shown in Table 1, may help to clinically recognize and support the detection and diagnosis of HIT.

There are 50 known non-synonymous single-nucleotide polymorphisms for the gene which codes DAO [24] and the histamine receptors [25]. These seem associated with a manifold of clinical symptoms and hundreds of symptom combinations [1]. Nonetheless, this may help to explain the extensive individual variability of HIT-related GI and extra-intestinal symptoms. Furthermore, this genetic variety in HIT might influence disease expression and individual responses to diets or treatment. However, the functional and clinical significance of these polymorphisms remains unknown. Another factor to consider is that histamine content in food is frequently unknown [26], and that it varies geographically depending on ripeness, storage time, and processing [6]. All of these variables need to be advised and may help to describe each person’s unique and occasionally changing tolerance level.

However, a diagnosis of HIT may be supported with a low serum DAO value <10 U/mL (normal >10 U/mL) [27]. GI symptoms, indicative of HIT, and a reduction of these when following a histamine-reduced diet, help to support a successful diagnosis [3]. Although, DAO values in serum are not established to correlate with DAO activity in the gut, a significant increase in serum DAO due to a strict histamine-reduced diet was demonstrated in patients with HIT [28]. Subsequently, a correlation between low DAO values and symptoms of HIT, and a response to histamine-reduced diet and/or to oral diamine oxidase supplementation was shown [29,30]. Some findings suggested that serum DAO values agree with symptoms of HIT [31]. In vitro and in vivo observations in mice, including administration of DAO into the gut lumen, were found to be a valid treatment for histamine-induced intestinal dysfunctions [32]. DAO made from the white pea (*Lathyrus sativus*) prevented histamine toxicity in vitro in human epithelial colorectal adenocarcinoma cells [33].

Currently, serum DAO is being determined using a radio extraction assay. However, this analytical method has limitations because only a relative amount of DAO in serum is quantified. With this method, a human DAO standard is not commercially available and absolute DAO quantities cannot be measured. Therefore, the development of new assay methods have shown an enzyme-linked immunosorbent assay, using human DAO as a standard, to be more accurate [34]. Nonetheless, the search for HIT diagnostic tests continues. 

## 4. Histamine Intolerance (HIT) Associated with Gastrointestinal (GI) Disorders

The various disorders and diseases, of which several are with unknown etiology, listed here are mainly presented with unexplained functional, nonspecific, non-allergic GI symptoms. Predominant symptoms are abdominal pain, bloating, and diarrhea. The GI symptoms include multiple individual combinations, which are also indicated by patients with HIT, with or without extra-intestinal symptoms.

However, DAO activity has been proposed as a marker of the integrity of intestinal mucosa. A recent study analyzed the molecular effects of histamine in human ileal neuroendocrine tumor cells, which are a model for gut enterochromaffin cells. The results indicated that enterochromaffin cells participate in intestinal intolerance or allergic reactions to food constituents associated with elevated histamine levels [35]. In inflammatory bowel diseases, reduced DAO activity was related to the degree of mucosal damage [36]. In Crohn’s disease, DAO was discussed as a marker for disease assessment [37]. Moreover, histamine content and secretion were found to be significantly increased in affected mucosa in Crohn’s disease and in ulcerative colitis [38]. Intestinal mucosa diamine oxidase activity was shown to reflect intestinal involvement in Crohn’s disease. Additionally, histamine was concluded to contribute to the mucosal reactions in the intestine and to reflect the degree of colonic inflammation [39]. Measurement of gut DAO activity was stated as a biologic marker of colorectal proliferation [40] and histamine catabolism was reported to be lower in the colonic mucosa of patients with colonic adenoma [41]. According to reports, in oncologic patients receiving chemotherapy, DAO activity may be a predictor of intestinal mucosal damage [42]. Serum DAO activity was reported to be a predictor of GI toxicity and malnutrition due to anticancer drugs [43]. Some results in children support the hypothesis of DAO being a marker of small intestinal functional integrity [44].

Histamine and DAO measurements, in vivo and in vitro, beyond any doubt need further investigation to shed light on the changes in gut tissues during various diseases and disorders.

### 4.1. Irritable Bowel Syndrome (IBS)-Like Disorders

Functional dyspepsia (FD), IBS, and small intestinal bacterial overgrowth (SIBO) are commonly reported, but solely symptom-oriented conditions. These clinical syndromes continue to be imprecise and were therefore re-named to “IBS-like” disorders [45]. They have no established pathophysiology, however, emerging evidence suggests that this paradigm may need revision [46]. In general, there is a lack of specificity of symptoms. Symptoms alone or symptom complexes can rarely, if ever, be used diagnostically. However, 80% of IBS patients identified food, including histamine, to be triggers for their GI symptoms [47]. Currently, IBS is defined using—and discussed within—the Rome IV criteria [48]. Furthermore, histamine was recently named to act as a key mediator in IBS [46]. FD is with absence of an organic disease a disorder with heterogeneous symptoms in the epigastric region. Patients with HIT report abdominal pain and postprandial fullness to be prominent symptoms and these are described in FD also as main symptoms. SIBO occurs with an abnormal increase in the overall bacterial population in the small intestine. Bloating and abdominal pain are primary symptoms in SIBO, very much comparable to HIT complaints [1]. However, there is an imprecise clinical overlap between IBS, FD, SIBO, and IBS-like disorders [49], and it is suspected that a number of different pathogenic mechanisms may be responsible. Dietary nickel (Ni) was described as a possible etiology for GI complaints in patients with IBS-like symptoms [50]. Remarkably, several foods described as containing Ni are also not well digested by HIT patients because of their high histamine content. 

Since there are symptom correlations in FD, IBS, and SIBO with HIT, determination of serum DAO values in well-defined patients may help to elucidate this link.

### 4.2. Non-Celiac Gluten Sensitivity (NCGS)

Due to all of these disorders being increasingly discussed in the media, a growing number of people are changing their diets. Currently, approximately 20% of the population are avoiding gluten in food because of a self-diagnosed, new clinical condition, referred to as non-celiac gluten sensitivity (NCGS). Unproven hypotheses of supposed health benefits due to gluten-free eating are wide-spread [51]. There are no medical tests available for the diagnosis of NCGS, and patients were also named “people without celiac disease avoiding gluten” [52]. Mainly, they are self-diagnosed and perform self-treatment with a gluten-free diet. It has been proposed that NCGS patients are a certain group of patients with IBS. Recently, a newly created term “non-celiac wheat sensitivity” has been suggested as a more accurate definition [53]. Moreover, widely used gluten-containing bakery goods and beers are prepared with histamine-producing yeast [54,55]. Additionally, foods like pasta, pizza, and bulgur are very commonly consumed with histamine-containing tomatoes [56] and seasonings. Generally, several gluten-free foods are low in histamine [57] and an increasing number of gluten-free products, including breads, are already labeled gluten-free and yeast-leavened [58]. Furthermore, we reported that in NCGS the GI and extra-intestinal symptoms resemble those found in HIT [59]. However, the reduction of gluten-containing food and drinks cuts the quantity of parallel consumed histamine. This may help to explain the current extraordinary popularity of gluten-free food. Lastly, the pathophysiology of these symptom-oriented disorders are unknown and HIT, with its plethora of symptoms, may play a role.

### 4.3. Food Intolerance/Malabsorption

There is growing public interest in adverse reactions to food that may influence and impair digestion. Food intolerance/malabsorption and people who experience GI reactions to food are reported to affect 20% of populations. Particularly the sugars (lactose and fructose), proteins (gluten), and biogenic amines (including histamine) are widely consumed and trigger GI reactions to food. However, quantity, type, and composition of dietary carbohydrates and proteins influence digestion and the metabolic output of gut microbiota [60]. In patients with HIT, a deranged gut flora and a change of the intestinal bacterial composition was demonstrated [61].

One or a combination of various food components cannot be degraded and/or absorbed properly within the gut. Combinations of LIT and fructose malabsorption were reported in >30% of patients with a carbohydrate intolerance/malabsorption [62]. In lactose-intolerant patients, the effect of HIT with different perceptions of functional, nonspecific, non-allergic GI symptoms was described [63]. During H_2_ breath tests of patients with LIT, the presence of additional fructose malabsorption and HIT significantly increased expiratory H_2_ values. This indicated that HIT embodies a separate GI disorder as food intolerance/malabsorption [64]. We described the fact that 55% of the patients with carbohydrate intolerance/malabsorption have serum DAO values below 10 U/mL as an indicator for HIT [5]. Further studies and descriptions of food intolerance/malabsorption, including the changes of the microbiota, in various combinations of intolerance/malabsorption, including HIT, are certainly needed.

### 4.4. Medications

Due to the increased release of histamine, or an inhibition of DAO, a variety of medications may influence HIT-related complaints or induce HIT [1,65,66] (Table 2). Approximately 20% of Europeans use DAO-inhibiting drugs on a regular basis, which significantly increases their susceptibility to adverse effects of ingested histamine [67]. Several of these drugs, including high-dose acetylsalicylic acid and nonsteroidal anti-inflammatory drugs, are available over the counter. However, they may cause GI side effects, including, with prolonged and high-dose use, an increased risk for GI bleedings [68]. Particularly, when exploring HIT-related symptoms, long-term treatments with these drugs need assessment, as well as reflection concerning the estimation of serum DAO values.

### 4.5. Disorders Associated with Mast Cells 

Mast cells are essentially involved in immunity and inflammation. Endogenous histamine is stored mainly in mast cells and basophils. The mast cells’ histamine release appears linked to GI-involving diseases like celiac disease (CD), eosinophilic gastroenteritis (EGE), and mast cell activation syndrome (MCAS) [4].

CD is a well-defined autoimmune disorder characterized by a gastrointestinal mucosal reaction to ingested gluten proteins. An overlap between CD and irritable bowel syndrome (IBS) exists and an improvement of symptoms with gluten restrictions in IBS was reported [69]. Mast cells releasing histamine and other inflammatory mediators have been linked with CD, showing that mast cells infiltrating the mucosa were associated with the severity of mucosal damage [70]. HIT was found in 55% of non-responsive CD patients and seems to play an important role in CD [71]. Originally, the Mas-related G protein-coupled receptor family (Mrgprs) members were identified as itch receptors in cutaneous sensory neurons. Now, they have become a target in abdominal pain research. Recently, a function for Mrgprs in the mouse gut nociceptive innervation and through Mrgprs elevation of histamine was found [72]. Given these functions of Mrgprs in humans, these findings deserve future research and may help to elucidate pathophysiology of HIT.

The underlying mechanisms of EGE are remain unknown. EGE is characterized by eosinophilic infiltration into the gastrointestinal mucosa. With the eosinophil–mast cell axis, which is involved in functional GI disorders, mast cells induce eosinophils and eosinophils activate mast cells in a co-dependent manner. Several symptoms in EGE, including GI complaints [73] resemble complaints in HIT [1]. It was suggested that cytokines combined with proliferation of eosinophils and histamine-releasing mast cells are involved in EGE [74]. The influence of a histamine-reduced diet on gastrointestinal symptoms in EGE remains to be discovered. 

Mast cell activation disorders have a certain relation of GI and extra-intestinal complaints with HIT. It was questioned whether HIT patients, some or all, could be seen within the mast cell activation disorders spectrum [75]. However, HIT mainly causes GI symptoms. A coexistence of the hypermobility spectrum disorder, Ehlers-Danlos syndrome, and the postural tachycardia syndrome exists [76]. Individuals with these syndromes have frequently functional GI disorders fulfilling criteria according to Rome IV [77] and present complex clinical challenges. The postural orthostatic tachycardia syndrome is associated with increased occurrence of IBS-like symptoms and is described according to abnormal increase in heart rate occurring after sitting or standing up. Future studies need to elucidate the pathophysiology of hypermobility spectrum disorders and postural orthostatic tachycardia, including evaluation of HIT in these syndromes. The role of histamine and a therapeutic effect of a histamine-reduced diet, and the effect of oral DAO capsules, remain to be determined.

### 4.6. Helicobacter Pylori (H.p.) Infection

In patients with unexplained functional GI symptoms, an *H.p.* infection needs to be considered. Accordingly, if it is detected, an eradication therapy is required [78]. Gastrin and histamine are the main stimulants of gastric acid secretion, and they, combined with *H.p.*, influence acid secretion [79]. Although the involvement of *H.p.* in functional dyspepsia is controversial, some studies from areas with a high prevalence of *H.p.* reported that eradication improves dyspepsia. Some association of *H.p.* infection and CD was reported because CD patients had significantly lower rates of gastric *H.p.* infection [80]. This argues that evaluations of GI infection with *H.p.,* the most prevalent human pathogen, with HIT are also needed.

## 5. Histamine Intolerance (HIT) Associated with Other Disorders

In various disorders and diseases, of which several are of unknown etiology, no clear link from headache and urticaria complaints to dyspepsia can be established. However, several patients in evaluations of described disorders were reported with unexplained functional, nonspecific, non-allergic GI symptoms. 

### 5.1. Headache

A specific pathophysiologic mechanism of headache is still unknown and migraines are defined as an untreatable disease [81]. Biogenic amines in wine, including histamine, tyramine, and putrescine [82], have relevance for headache and histamine-related symptoms [83]. Red wines contain clearly more than double the histamine concentrations with >2200 µg/L, compared to white wines (~900 µg/L histamine) [6]. An increased risk for migraines was demonstrated in patients with some DAO genotypes and allelic variants [84]. A high incidence of DAO deficiency at nearly 90% was observed in migraine patients [85]. Recently, a study demonstrated that oral ingestion of capsules with DAO significantly reduces headaches in migraine patients [86]. Subsequently it was shown that headaches in HIT patients—as one of the many symptoms in HIT—were considerably reduced due to oral DAO supplementation [30]. The organic wine industry was reported to lower the amount of histamine in wine. This decreases headaches and other adverse effects usually provoked by drinking wine [87]. Associations between migraine, celiac disease, non-celiac gluten sensitivity, and low activity of DAO were hypothesized. It was stated that patients with low serum DAO values were more severely impacted by migraine than healthy persons with normal DAO activity [88]. Bloating with headache was mentioned by 63% of HIT patients as one of the most common HIT-related symptom combinations [1]. Thus, the histamine content of consumed food may play a key role in triggering migraines and headaches, and there are certainly additional evaluations necessary.

### 5.2. Urticaria

Urticaria is a relapsing-remitting skin disease with unknown etiology. The majority of cases are called idiopathic or autoreactive. It has a significant impact on affected patients and reduces their quality of life. The mainstay of medical therapy is symptom management, including the use of H1 antihistamines. Histamine-releasing mast cells and basophils are known as primary inflammatory cells in urticaria. Degranulation and release of vasoactive substances, including histamine and other pro-inflammatory mediators, causes vasodilation, sensory nerve activation, plasma extravasation, dermal edema, and wheals. Although further evaluations are needed, it appears that some patients have dietary triggers contributing to this skin disease and a number of food ingredients have been reported to worsen symptoms [89]. In patients suffering from urticaria, accompanied by functional GI symptoms, a histamine-reduced diet was demonstrated therapeutically useful, simple, and economically efficient. The diet decreased symptoms and increased quality of life in urticaria patients [90]. Additionally, oral DAO supplementation was found to improve the degree of urticarial activity score, which was inversely correlated with the levels of serum DAO [91]. Dietary intervention in urticaria patients with histamine-reduced diet and/or oral DAO capsules obviously needs additional investigations.

### 5.3. Further Diseases and Disorders

Occurrence of HIT, causing GI complaints, was described in some patients with rare diseases, such as primary epiploic appendagitis, beta-thalassemias minor, and Gullo syndrome [92]. However, it has also has been reported in several other cases and further groups of patients with HIT-related symptoms. In all of these examples, the reduction of histamine in food with initiation of a histamine-reduced diet was helpful to decrease complaints.

Recently, a patient with already performed Nissen fundoplication had recurring gastroesophageal reflux symptoms with coughing and increased throat clearing. Lung function and gastroesophageal reflux disease were tested but not found as the cause of these symptoms. After a successful diagnosis of HIT, a histamine-reduced diet resulted in apparent improvement of the patient’s complaints [93].

In a female patient with anorexia nervosa, HIT-related symptoms arose. After she followed a histamine-reduced diet, she experienced weight gain and improvement of GI symptoms [94]. 

Drinking wine with high histamine content caused coughing and wheezing with bronchoconstriction in patients with HIT [95].

Etiology and triggers of atopic dermatitis are still under debate. The treatment of atopic dermatitis, in a patient who also had HIT-related GI symptoms, was successful with a histamine-reduced diet [96].

Interestingly, histamine secretion from bacteria within the gut of mice was shown to have immunological consequences within the lung [97]. On the other hand, a pilot study reported on a histamine-reduced diet, which might have had an active and direct impact on asthma symptoms in children, without known HIT [98].

Somewhat conflicting results are reported on DAO and histamine values in patients with multiple sclerosis (MS), an autoimmune disease of the central nervous system. Incompatible low DAO and low histamine serum levels have been documented in MS [99]. However, the pathogenesis and an association to HIT are unknown, but this again documents the need for further investigations.

Usually, GI endoscopy with histologic evaluation of GI mucosa and radiological evaluation, including ultrasound, of the abdomen are valuable additional options for examination of patients, especially aged >55 years, with functional, nonspecific, non-allergic GI symptoms [100]. If signs of GI infection are evident, then specific anti-microbial or anti-parasitic treatments, that may reverse the disease, need consideration.

## 6. Discussion

Generally, the growing number of scientific studies has led to a better understanding of the role of food ingredients causing GI complaints. If biogenic amines, including histamine, cannot be absorbed and digested properly in HIT then this causes non-allergic, functional, nonspecific GI and extra-intestinal complaints. GI bacteria use catabolic enzymes to degrade and ferment carbohydrates and proteins from ingested food [101], as well as in HIT, a changed intestinal bacterial composition was reported.

Although there is limited availability of sufficiently sensitive and specific diagnostic procedures, it seems essential to assess HIT individually with the currently available methods. The thorough anamnesis of all HIT-linked complaints is the mainstay of HIT diagnosis. However, for evaluation of patients with symptoms, suspected to have HIT, the questionnaire indicated here (Table 1) may be helpful. Medical personnel should be aware that a large number of HIT patients have, to a certain extent, GI complaints. However, distinct patients may present only single symptoms like, e.g., postprandial vertigo or headache. Additionally, several studies demonstrate the growing importance of serum DAO determinations. So far, some studies conducted report on an insufficient low number of patients. This demonstrates the need for further evaluations of HIT in clearly defined and larger patient groups. 

After detailed diagnosis of HIT, the GI complaints and extra-intestinal symptoms can be decreased by reducing the consumption of histamine and/or oral DAO supplementation. Capsules containing DAO seem helpful but there is an ongoing discussion about the therapeutically helpful dosage [102]. Each patient’s tolerance level should also be considered, when recommending dietary restrictions for long-term symptom reduction. Nonetheless, the permanent compliance with a locally-available, histamine-reduced diet, is challenging for patients. Certainly, this represents the future need of the histamine content indication on food labels. Improvements and developments of new methods for the determination of histamine and biogenic amines in food will allow this [103,104,105].

A strong association of histamine and HIT with functional GI complaints in various disorders and diseases is assumed, several of these with a so-far unknown origin. Therapeutically helpful, in many ways, is the reduction of the food histamine load. However, if HIT is present, an experienced, a registered dietician may help to design an individually tailored diet. The reduction of ingested histamine with a histamine-reduced diet is helpful for HIT-related symptoms and available at low cost. This seems to positively influence the pathophysiologic process, including inflammation, and accomplish symptom reduction. The oral supplementation with DAO capsules represents an additional therapeutic option to lower HIT-related symptoms. So far, an agreement for dosing of oral DAO supplements has not been established. Commercially available capsules contain 4.2 mg extracted pig kidney proteins with 0.3 mg DAO. These are suggested up to three times per day before meals. Although, it was reported that a higher activity of DAO seems required for a satisfactory histamine degradation [102]. The dietary intervention with histamine-reduced diet and/or oral supplementation of DAO clearly needs additional investigations. Interdisciplinary management is necessary, so that all etiologic and therapeutic possibilities are included in the evaluation of patients with HIT [4,106]. The diagnosis of HIT helps patients to put their symptoms into context, to reduce complaints and to improve their quality of life. This shows why additional scientific evaluations in these demonstrated diseases and disorders are certainly necessary. Nonetheless, the development of new laboratory methods will improve determination of DAO and histamine in serum, and histamine in food.

## 7. Conclusions

In conclusion, HIT seems to play a more significant role in GI disorders and complaints, and in several extra-intestinal disorders, than so far anticipated. Overall, scientific evidence of histamine involvement and HIT in the demonstrated diseases and disorders is scarce but increasing. A detailed diagnosis of HIT, with currently available methods and tests, including the questionnaire (Table 2), is helpful and necessary to evaluate each patient with HIT-related GI and extra-intestinal complaints. With the increasing scientific evidence, we demonstrate, that HIT is primarily a GI disorder and that it originates in the gut.

## Figures and Tables

**Table 1 nutrients-13-01262-t001:** Example of a standardized questionnaire for evaluation of patients with suspected histamine intolerance (HIT).

Severity of Complaints: No Symptoms (0), Mild (1) to Very Severe Complaints (5)						
	**Gastrointestinal**					
	0	1	2	3	4	5
Abdominal pain	o	o	o	o	o	o
Intestinal colics	o	o	o	o	o	o
Bloating	o	o	o	o	o	o
Diarrhea	o	o	o	o	o	o
Constipation	o	o	o	o	o	o
Nausea	o	o	o	o	o	o
Belching	o	o	o	o	o	o
Vomiting	o	o	o	o	o	o
Postprandial fullness	o	o	o	o	o	o
Menstrual cramps	o	o	o	o	o	o
	**Skin**					
	0	1	2	3	4	5
Pruritus	o	o	o	o	o	o
Eczema	o	o	o	o	o	o
Reddened skin	o	o	o	o	o	o
Swollen, reddened eye lids	o	o	o	o	o	o
	**Cardiovascular**					
	0	1	2	3	4	5
Headache	o	o	o	o	o	o
Dizziness	o	o	o	o	o	o
Hypotonia	o	o	o	o	o	o
Palpitations	o	o	o	o	o	o
Collapse	o	o	o	o	o	o
	**Respiration**					
	0	1	2	3	4	5
Rhinorrhea	o	o	o	o	o	o
Nose congestion	o	o	o	o	o	o
Sneezing	o	o	o	o	o	o
Asthma	o	o	o	o	o	o
	**Additional Complaints (Please List Symptoms that Have Not Yet Been Listed)**					
	0	1	2	3	4	5
	o	o	o	o	o	o
	o	o	o	o	o	o
	o	o	o	o	o	o

Please tick complaints which you experience mainly after ingestion of food. Adapted according to reference [1].

**Table 2 nutrients-13-01262-t002:** Medications which may influence diamine oxidase and/or histamine.

Medications	Generic Name
Analgesics	Acetylsalicylic acid, Metamizole, Morphines, Nonsteroidal anti-inflammatory drugs, Pethidine
Antiarrhythmics	Propafenon
Antibiotics	Cefuroxime, Cefotiam, Isoniazid, Pentamidine, Clavulanic acid, Chloroquine
Antidepressants	Amitriptylline
Antifungal	Pentamidine
Antihypertensives	Verapamil, Alprenolol, Dihydralazine
Antihypotensives	Dobutamine
Antimalarial	Chloroquine
Broncholytics	Aminophylline
Cytostatics	Cyclophosphamide
Diuretics	Amiloride
H2 receptor antagonists	Cimetidine
Local anesthetics	Prilocaine
Motility agents	Metoclopramide
Mucolytics	Acetylcysteine, Ambroxol
Muscle relaxants	Pancuronium, Alcuronium, D-Tubocurarin
Narcotics	Thiopental
Radiological contrast media	
Vitamines	Ascorbic acid, Thiamine

Adapted according to references [2,62].

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
