# Peer review of "Histamine Intolerance Originates in the Gut"

_nutrients, 2021, doi:10.3390/nu13041262_

Round 1

Reviewer 1 Report

The review entitled Histamine intolerance originates in the gut is a good work dealing with histamine's effects exerting a plethora of gastrointestinal symptoms on susceptible individuals. I think that this work has enough quality to be published on Nutrients after a minor revision.

Some suggestions:

  1. I suggest reconsidering some outdated references (before 2000) if feasible, replacing them with the latest ones.
  2. Page 2, line 51: Actually, tyramine is another vital biogenic amine, often related to headache and increased blood pressure. Authors should at least mention it, possibly disregarding agmatine. Authors should consider tyramine as a headache cofactor also in section 5.1.
  3. Page 2 lines 66-70 and Page 8 lines 328-343: Whilst, low amount of histamine does not exert histamine intolerance symptoms, according to some studies, the effects of histamine could be enhanced by the co-occurrence of other biogenic amines (like putrescine) and alcohol. I suggest to see the following study: Esposito, F., Montuori, P., Schettino, M., Velotto, S., Stasi, T., Romano, R., & Cirillo, T. (2019). Level of biogenic amines in red and white wines, dietary exposure, and histamine-mediated symptoms upon wine ingestion. Molecules, 24(19), 3629.

Reviewer 2 Report

The authors have done a good job in writing a comprehensive manuscript.  Congratulations.

Reviewer 3 Report

The authors do a fantastic job in this review and it is well-written. Please consider the following edits that will need to be made and this article is otherwise suitable for publication: 

1) In the section "4.1. Irritable bowel syndrome (IBS)-like disorders": Please expand a bit more on how each of the conditions like FD, SIBO, and IBS actually correlate with histamine sensitivity. I saw FD and SIBO mentioned without much discussion in relation to histamine. 

2) The authors note in 4.2, "However, the reduction of gluten-containing food and drinks cuts the quantity of parallel consumed histamine. This may help to explain the current extraordinary popularity of gluten-free food"...Please provide a better explanation as to why gluten intake reduction cuts the quantity of consumed histamine by providing more description and discussion of science underlying this comment. 

3) The authors note in 4.4 "However, these are known to frequently cause a variety, of, potentially life-threatening, GI side effects [66]." Please provide more context for this statement. Are the authors referring to increase risk for GI bleeds with NSAIDS and increased liver toxicity with acetominophen as that seems a bit of a stretch and depends on how often the Europeans are using them and in what does. Please offer more of an explanation and consider deleting this sentence as it does not add much and may actually be overcalling the risk here since most do not simply take mega doses. In general these medications are NOT life threatening and only when taken in large quantities or with prolonged use. For that reason, this sentence appears to misrepresent these medication risks. 

4) In section 5.1 the authors state "The low amount of histamine in wine, from organic wine industry, decreases headaches and other adverse effects usually provoked by drinking wine [85]." This seems to contradict the previous sentences and please reconcile. 

5) In the section 5.3. Further diseases and disorders, the authors should separate each new disease discussion (even is just reporting cases) into a new paragraph. Otherwise, it is hard to follow everything when it is all put together in one long paragraph. Also, the comments on soy sauce are not convincing and a stretch and please delete these sections ("A cheilitis, characterized by inflammation of the lips, was found to be caused by the ingestion of soy sauce [92]. However, after fermentation, soy sauce is known for having a high content of histamine and biogenic amines [93].")

5) Please include a table on the typical dosing of DAO so that readers can have context to how the DAO can be dosed. 
